

# Comparison of EfficientNet CNN models for multi-label chest X-ray disease diagnosis

Murat Ucan[1], Buket Kaya[2], Osman Aygun[3], Mehmet Kaya[4] and Reda Alhajj[5]

[1] Computer Technologies, Dicle (Tirgris) University, Diyarbakir, Turkey
[2] Electronics and Automation, Firat (Euphrates) University, Elazig, Turkey
[3] Keban Vocational School, Firat (Euphrates) University, Elazig, Turkey
[4] Computer Engineering, Firat (Euphrates) University, Elazig, Turkey
[5] Computer Science, University of Calgary, Calgary, AB, Canada

Corresponding authors
Mehmet Kaya, kaya@firat.edu.tr
Reda Alhajj, alhajj@ucalgary.ca

## ABSTRACT

The analysis of chest X-ray images, which are critical for the early diagnosis of many diseases, is a difficult and time-consuming process due to the multiple labeling requirements and similar looking pathologies. In traditional methods, expert physicians analyze high-resolution chest X-ray images to diagnose these diseases using observational methods, a process that can lead to human error and hence misdiagnosis or underdiagnosis. In this study, we aim to autonomously detect 14 different diseases that significantly affect human health and some cases even lead to death using chest X-ray images in a multi-class manner using deep learning techniques. Previous studies on chest X-ray images focus on a single disease or have low success rates, and the architectures presented in previous studies have high computational costs. The novelty of this work is that it presents a hybrid lightweight, fast and attention-based architecture with high classification performance. In this study, we used the ChestX-Ray14 dataset consisting of 112,104 labeled chest X-ray images of 14 disease classes. Eight deep learning architectures (EfficientNetB0-B7) and coordinate attention mechanism are used in the training and testing processes. The proposed EfficientNetB7 architecture achieved an average overall classification performance with an AUC value of 0.8265. The EfficientNet enhanced with coordinate attention architecture achieved a classification success with an AUC value of 0.8309. Moreover, when the proposed architecture and the individual disease classes are considered separately, higher classification success is achieved for eight of the 14 diseases in the dataset. Finally, the results of this study outperformed the classification performance of other similar studies in the literature in terms of AUC score. The results obtained in our study show that the proposed deep learning based lightweight and fast architecture can support radiologists in decision making in disease diagnosis. The use of autonomous disease diagnosis systems can support the protection of human health by preventing incomplete or erroneous diagnoses.

# INTRODUCTION

Chest X-rays are important medical images that allow the diagnosis of many diseases and are widely used by specialized physicians (*Albahli et al., 2021*). Many diseases such as atelectasis, cardiomegaly, consolidation, edema, effusion, emphysema, fibrosis, hernia, infiltration, mass, nodule, pleural thinning, pneumonia and pneumothorax can be diagnosed from chest X-ray images (*Wang et al., 2017*). In traditional methods, diseases are detected by manual examination of chest X-ray images by specialized doctors. However, the traditional diagnosis phase is slow and prone to human error as it contains findings of many diseases (*Ucan et al., 2025*). In addition, an expert doctor is needed to diagnose diseases by examining chest X-ray images. In hospitals where there is no specialized doctor who can interpret chest X-ray images, primary care or necessary critical referrals cannot be made to patients by interpreting the images. In addition, existing AI-based research in this area of study does not have high enough diagnostic success. For these reasons, developing systems that can autonomously diagnose diseases from chest X-ray images and support doctors in decision making is an important problem that researchers should focus on.

In recent years, autonomous disease diagnosis from many medical images such as brain MRI (*Ucan, Ucan & Kaya, 2023*), gastroenterology (*UCan, Kaya & Kaya, 2022*), EEG (*Dişli et al., 2025*), CT (*Gulsoy & Baykal Kablan, 2025*) has been studied using deep learning architectures. The high achievements obtained in different image inputs shed light on the researchers for future studies. There are also some studies conducted by researchers on chest X-ray images. However, these studies do not have high enough success rates to be used in hospitals in the real world. They also have long training times and high computational costs as a result of studies focusing on high success rates. There is a need to develop new methods with low computational cost and high success rates that can be used in hospitals (*Rehman et al., 2021*).

Speed, high diagnostic success and architectural complexity are the most important factors in deep learning-based disease diagnosis studies. Considering these factors, researchers choose a deep learning architecture. EfficientNet deep learning architecture is a CNN-based architecture used by many researchers and focuses on achieving high success with low computational cost (*Goutham et al., 2022*). In addition, EfficientNet offers researchers the opportunity to train their models with shorter training times with the advantage of low number of parameters. These advantages are important variables for the choice of architecture in deep learning based systems where medical images are used as input. Diseases such as SARS, COVID-19 (*Kumar et al., 2021*) and M-Pox (*Vuran et al., 2025*), which spread rapidly and affected masses worldwide, have taught humanity the importance of rapid diagnosis and achieving high success with little data. EfficientNet is a deep learning architecture suitable for use on medical images due to its low computational cost and high performance.

In this study, using chest X-ray images in the ChestX-ray14 dataset; atelectasis, cardiomegaly, effusion, infiltration, mass, nodule, pneumonia, pneumothorax, consolidation, edema, emphysema, fibrosis, PT, hernia diseases were detected using convolutional neural networks (ESA) methods. Chest X-ray images were trained using

eight architectures in total, EfficientNetB0-B7 architectures. The performance comparison and classification successes of the architectures used were analyzed in detail and given with graphics. The proposed study aims to develop an autonomous deep learning-based system that is suitable for mobile use and has low computational complexity, which can support specialist doctors trying to diagnose diseases on chest X-ray images. The topics on which the study focuses and will contribute to the literature are presented below:

- While large-scale deep learning architectures focus on higher performance, they ignore computation time and architectural complexity. In our study, a framework is presented by comparing architectures that can be used easily even in mobile systems and can achieve high success despite having less computational complexity.
- The study finds a solution to the multi-class classification problem by applying EfficientNetB0-B7 and MobileNet architectures to chest X-ray images.
- The study sought a solution to which architecture could be used effectively in mobile devices in hospitals in rural areas where there are no specialist doctors who can interpret chest X-ray images and there are relatively less infrastructure facilities.
- The use of deep learning models suggested by the study can provide faster disease detection compared to traditional methods.
- Using the proposed architecture during the diagnosis of diseases can help reduce misdiagnoses and treatments caused by human errors.

## LITERATURE REVIEW

With the increase in the computational capacity of computers and the increasing confidence in artificial intelligence applications, the use of artificial intelligence in the analysis of medical images is also increasing. Deep learning architectures, a current subfield of artificial intelligence, have also achieved promising results in medical image analysis. Researchers have conducted many studies that can diagnose diseases from chest X-rays using deep learning architectures. The authors performed disease detection on the ChestX-ray14 dataset using deep learning methods in _Rajpurkar et al. (2017)_. The ChestX-ray14 dataset is a chest X-ray dataset created by the authors. The images in the dataset were sized to $224 \times 224$ and used in deep learning architecture. They used the deep learning architecture they developed, called CheXNet. CheXNet is a 121-layer dense convolutional neural network based model. To test the effectiveness of their study, the authors gave the same test pictures to four expert radiologists and their own model and compared the results. The mean F1 score of the radiologists was calculated as 0.387. In the ChexNet model, the F1 score was calculated as 0.435. The results show that the ChexNet model detects disease better on average than specialist radiologists.

The authors detected abnormalities on chest X-ray images using the PLCO and ChestX-ray14 datasets in _Gündel et al. (2019)_. The ChestX-ray dataset contains images of 14 diseases. The PLCO data set contains data on 12 diseases. They trained the datasets with the prepared architecture using the DenseNet-121-based convolutional neural network architecture. The authors obtained a mean value of 0.865 AUC in 12 diseases in the PLCO dataset with the DNet architecture they developed. In the ChestX-ray14 dataset, they obtained

an average of 0.807 AUC in 14 diseases. Performed disease detection from chest X-ray images using the ChestX-ray8 dataset prepared at hospital scales in *Wang et al. (2017)*. The ChestX-ray8 dataset contains a total of 108,948 front view chest radiology images from 32,717 patients. There are data on 8 different diseases in the data set; these are atelectasis, cardiomegaly, effusion, infiltration, mass, nodule, pneumonia and pneumothorax. The authors achieved 63.9% success in AlexNet architecture, 63.9% in GoogLeNet architecture, 62.5% in VGGNet-16 architecture, and finally 69.6% in ResNet-50 architecture.

The authors studied disease detection from chest X-ray images using the ChestX-ray14 dataset and the PyTorch library in *Wang et al. (2020a)*. The authors developed a new architecture by using the Dense block consisting of BN, ReLu, Conv(1x1), BN, ReLu, Conv(3x3) layers four times in a row and adding transition blocks between the dense blocks, and used this architecture in their work. The authors obtained an average of 0.53 AUC with the SVM architecture, 0.80 with the LSTM architecture, and 0.74 with the DCNN architecture. In the architecture they developed, the authors obtained an average of 0.82 AUC in 14 diseases. The authors developed a new model with convolutional neural networks using the triple attention mechanism in *Wang et al. (2021)*. They performed training and testing using the ChestX-ray14 dataset. In the training phase of the architecture they developed, they used the weights of the pre-trained ImageNet dataset. The authors shared the results of training and testing processes in 6 different attention mechanisms within the scope of the study. A3Net, which is the most successful convolutional neural network with attention mechanism, which the authors recommend to use, achieved an average value of 0.826 AUC.

*Li et al. (2021)* developed a lesion sensitive convolutional neural network to classify chest radiology images. Using the ChestX-ray14 data set in the study, the classification of 14 diseases was carried out with the developed model. In the developed model, lesion attention layer and residual blocks are used. When the test successes were examined, they obtained an average value of 0.888 AUC in the data of 14 diseases. The authors proposed an attention-based convolutional neural network model based on disease categories in *Guan & Huang (2020)*. ChestX-ray14 data set was used in the study and disease detection of 14 diseases was made. The attention mechanism used in the study was designed using three convolution layers, two ReLU activation layers and one sigmoid activation layer. The authors obtained an average of 0.810 AUC in 14 diseases with the model they developed using ResNet-50, and 0.816 AUC with the model they developed using DenseNet-121. *Chen et al. (2020)*, presented a two-stream collaborative architecture by segmenting chest X-ray images and then applying classification. In the study, first of all, a lung mask was created from chest X-rays using the U-Net architecture. The images cut by masking with the lung image and U-Net architecture without preprocessing were used together in the two-stream architecture. The authors obtained a mean value of 0.823 AUC in 14 diseases with the TSCN architecture they developed.

Another study prepared using the ChestX-ray14 dataset was conducted by *Nawaz et al. (2023)* to classify chest X-ray diseases. The researchers used the number of diseases in the data set as 8, based on expert opinion. In the last case, they carried out a classification study of atelectasis, cardiomegaly, effusion, infiltration, mass, nodule, pneumonia and

pneumothorax diseases. The researchers obtained an average AUC evaluation metric score of 0.908. Another important study using chest X-ray images was conducted by *Hamza et al. (2022c)*, focusing on covid disease. Using ResNet50 and InceptionV3 deep learning architectures, COVID-19 disease was classified using chest X-ray images. In the study, accuracy between 99.6% and 100% was achieved in five data sets. In another study *Hamza et al. (2022a)* trying to detect COVID-19 disease using chest X-ray images, 99.4% classification success was achieved using EfficientNet architecture and Grad-CAM. Similarly, in another study *Hamza et al. (2022b)* where it was suggested to use CNN and LSTM architectures together, 98% success was achieved in the proposed tri-layered neural study.

In another study focusing on improving the success of autonomous disease diagnosis using chest X-ray images, distillation techniques were used (*Ho & Gwak, 2020*). Integrating distillation techniques into the popular deep learning architectures ResNet-152 and DenseNet-121, the study examined the effect of information distillation on classification success. Another important improvement in CNN-based architectures that increases the classification success is the use of attention layer. CNN architectures and attention layers have been used together in some studies aiming for higher success for multi-class classification of chest X-ray images. *Wang et al. (2021)* used the triple attention mechanism with CNN architectures and observed its effect on classification success.

There are also some studies in the literature that focus only on a specific disease in the diagnosis of chest diseases and diagnose the related disease with artificial intelligence-based autonomous systems. In a study focusing on disease diagnosis from chest images, an AI-based solution was presented on tuberculosis (*Duong et al., 2021*). In the study on EfficientNet, Vision Transformer and hybrid versions of these architectures, the researchers achieved 97.72% accuracy. *Duong et al. (2023)* detected the early diagnosis of COVID-19 disease with artificial intelligence-based systems using chest X-ray and lung computed tomography images. As a result of the experiments using architectures such as EfficientNet and MixNet and different transfer learning approaches, the researchers achieved over 95% accuracy.

Existing studies in the literature focusing on disease detection from chest X-ray images have significant limitations such as computational complexity, low diagnostic accuracy and use in real-world problems. Traditional deep learning architectures have much higher computational complexity in the training process due to the high number of parameters. This makes it difficult to work with large data sets, resulting in long training times. In addition, studies focusing on disease detection from chest X-ray images have not yet achieved acceptably high success rates. These limitations lead to the conclusion that models with high computational complexity are not suitable for practical use in clinics. Considering the limitations of existing work, we chose to use the EfficientNet architecture in this study to optimize the accuracy-performance trade-off and achieve higher accuracy in the detection of fourteen diseases from chest X-ray images. EfficientNet model scaling ensures high classification performance and low computational complexity. By leveraging EfficientNetB0-B7, this study aims to develop a model that not only achieves

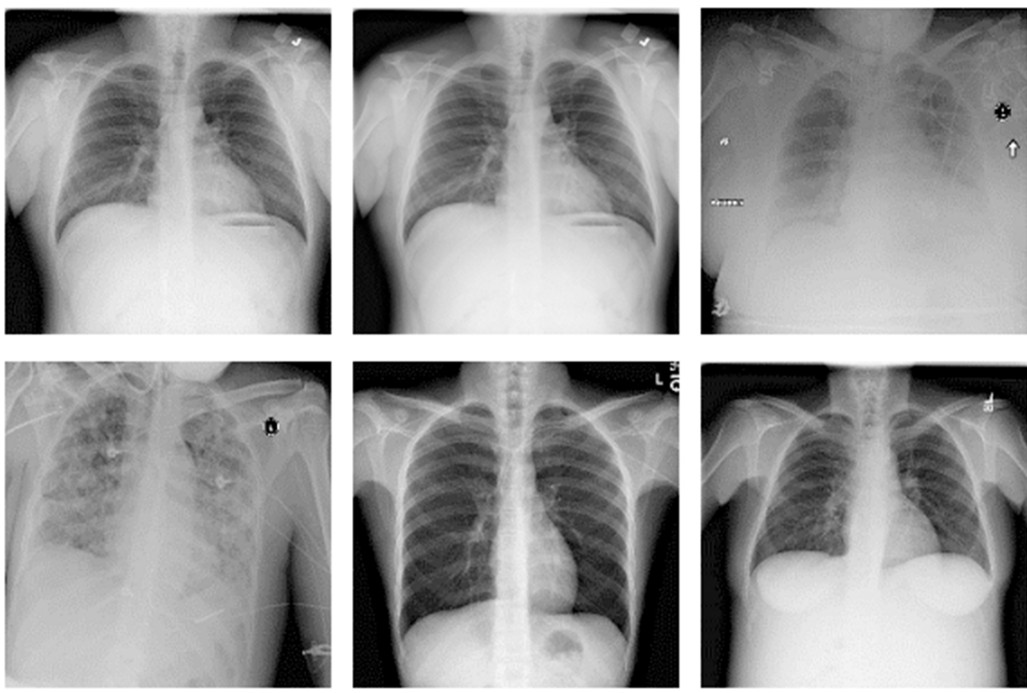

**Figure 1** **ChestX-ray14 dataset sample images.**

high classification accuracy but also provides practical applicability in real-world clinical settings.

## MATERIAL AND METHODS

In the study, a deep learning model will be developed that can detect disease using chest X-ray images and the results of the experiments will be given in detail. In this section, information will be given about the characteristics of the data set used in the training, testing and validation stages, the training environment and parameters used in the classification will be given, and detailed information will be given about the deep learning model that is used.

### Dataset

Training, validation and testing processes of our study were carried out using the ChestX-ray14 dataset, one of the largest publicly available datasets containing chest X-ray images (*Wang et al., 2017*; *National Institutes of Health Clinical Center, 2024*). The dataset contains a total of 112,120 chest X-ray images of 30,805 different patients. There is at least one chest X-ray image of one patient. But there are also patients with multiple chest X-ray images. Chest X-ray images of some patients taken from five different angles are available in the dataset. Some randomly selected images from the dataset of chest X-ray images are given in Fig. 1.

The dataset collected in accordance with the multi-class classification problem consists of images of 14 different diseases that can be detected from chest X-ray images and the

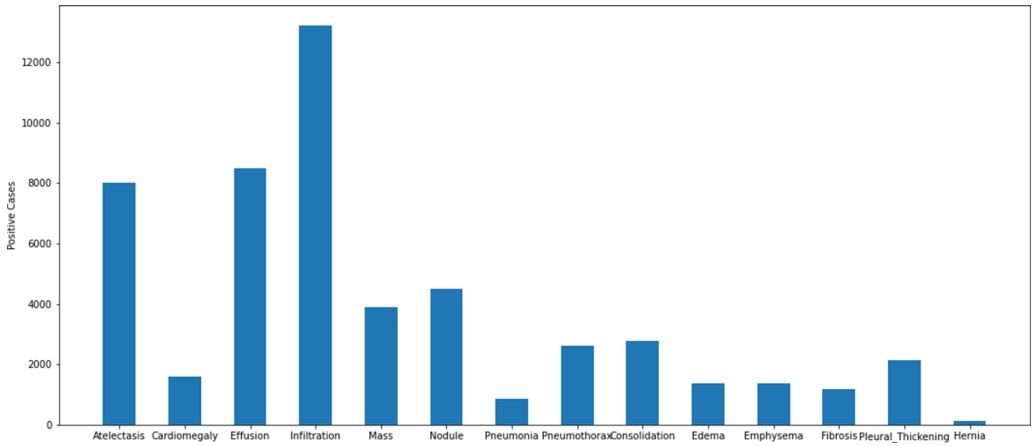

**Figure 2** Distribution of disease classes in the ChestX-ray14 dataset.

labels of these images. The diseases found in the data series are as follows: atelectasis, cardiomegaly, consolidation, edema, effusion, emphysema, fibrosis, hernia, infiltration, mass, nodule, pleural thickening, pneumonia, pneumothorax. The images in the dataset are 8-bit black and white images. There is not a uniform distribution of the photographs within the data set in accordance with the disorders. There are different numbers of images of different diseases. A graphical representation of the distribution of the data set that was used in the study according to the different disease groups can be found in Fig. 2.

Some of the chest X-ray images in the dataset belong to individuals diagnosed with more than one disease. Due to this situation, as in real-life problems, each image in the dataset is associated with more than one disease label. In the preprocessing stage, a publicly available CSV file was used with the dataset to manage multiclassing. This file is a data matching tool that contains image names and disease names. A precise and careful matching of disease labels and images was performed prior to the architecture training and testing in the study. In the next stage of the preprocessing step, the size of the images was harmonized with the input requirements of the deep learning architectures under study. At this stage, the images were resized to the expected input sizes for all EfficientNet architectures. Finally, since the chest X-ray images are in grayscale format, they were expanded from a single channel to a three-channel structure. In this way, EfficientNet models, which are traditionally trained on RGB, were adapted to the standard input format.

Some images in the dataset are labelled to contain multiple diseases. An example image in the dataset reported that the patient with which it was associated had atelectasis, consolidation, effusion, emphysema, mass, and pneumothorax diseases simultaneously. Although it is possible to have findings of 6 diseases at the same time in a patient, it is not possible to train with deep learning algorithms with low number of support data. There is only one image of the patient who has six diseases at the same time. This makes it impossible to train using the relevant image. For this reason, training, validation and testing subsets were created by taking the support value as at least 12 during the matching

**Table 1   Comparison of EfficientNet and other popular deep learning architectures.**

| Model | Size (MB) | Parameters | Depth | Time (ms) per inference step (GPU) |
|---|---|---|---|---|
| ResNet50 | 98 | 25.6M | 107 | 4.6 |
| ResNet101 | 171 | 44.7M | 209 | 5.2 |
| ResNet152 | 232 | 60.4M | 311 | 6.5 |
| DenseNet121 | 33 | 8.1M | 242 | 5.4 |
| DenseNet169 | 57 | 14.3M | 338 | 6.3 |
| DenseNet201 | 80 | 20.2M | 402 | 6.7 |
| MobileNet | 16 | 4.3M | 55 | 3.4 |
| EfficientNetB0 | 29 | 5.3M | 132 | 4.9 |
| EfficientNetB1 | 31 | 7.9M | 186 | 5.6 |
| EfficientNetB2 | 36 | 9.2M | 186 | 6.5 |
| EfficientNetB3 | 48 | 12.3M | 210 | 8.8 |
| EfficientNetB4 | 75 | 19.5M | 258 | 15.1 |
| EfficientNetB5 | 118 | 30.6M | 312 | 25.3 |
| EfficientNetB6 | 166 | 43.3M | 360 | 40.4 |
| EfficientNetB7 | 256 | 66.7M | 438 | 61.6 |

of the diseases and images in the data set. The graph containing the disease distributions in the data set shared in Fig. 2 was also obtained using a similar approach.

## EfficientNet

The EfficientNet model is an architecture used by many researchers in the literature developed in 2020. The efficient architecture achieved 0.844 accuracy in classifying the ImageNet dataset (*Tan & Le, 2019*). The most important feature that distinguishes EfficientNet architectures from other architectures is that it carries out training operations using fewer parameters. The EfficientNet-B7 architecture makes calculations using 66M parameters while classifying the ImageNet dataset (*Tan & Le, 2019*). Other deep learning architectures popular in the literature are DenseNet and ResNet architectures. The parameter counts and depth comparison of the EfficientNet architecture and other popular architectures are given in Table 1. The depth shown in Table 1 is the topological depth of the deep learning network. In other words, depth is the sum of the number of layers such as convolution, normalization, activation in the designed architecture.

EfficientNet architectures consist of eight different convolutional neural network models between B0 and B7. EfficientNet architecture uses compound scaling to improve classification performance by scaling depth, width and input image resolution in a balanced way. While traditional CNN-based deep learning architectures can scale only one of these components, the EfficientNet model family optimizes all three components within itself. Depth scaling allows learning complex features by adding more convolutional blocks. Width scaling increases the number of filters in the layers, allowing more detail to be captured. Resolution scaling increases the size of the input image, allowing the model to learn finer details. This approach ensures that the model is balanced in terms of efficiency

and classification success. ResNet50 architecture, which is the basic model of ResNet, uses 25.6M parameters (*Cinar, Kaya & Kaya, 2022*). 8.1M parameters are used in DenseNet121 architecture, which is the basic model of DenseNet (*Minaee et al., 2020*). Compared to these, only 5.3M parameters are used in the EfficientNetB0 architecture, which is the basic architecture of the EfficientNet architecture (*Vijayalata et al., 2022*).

In terms of depth, ResNet50 has 107 layers, DenseNet121 reaches 242 layers, and EfficientNetB0 consists of 132 layers. This shows that the EfficientNet architecture has similar depth to the ResNet architecture, but less depth than the DenseNet architecture (*Wang et al., 2020b*). Although it has similar depth with the ResNet architecture, When compared to the Resnet-50 architecture, the EfficientNetB0 architecture makes use of 4.9 significantly fewer parameters than the ResNet-50 architecture does. The reduction in the number of parameters significantly reduces the training time of the architecture (*Ataş et al., 2022*).

When we examine the time performance of the models, we see that the ResNet50 and EfficientNetB0 models complete their training processes at similar times. When we compare the EfficientNetB0 model and the EfficientNetB7 models, it is seen that there is a 12-fold difference between them. In other words, it takes 12 times more time to train the EfficientNetB7 model than the EfficientNetB0 model. The data and explanations in the table show that the EfficientNet deep learning architecture has a better balance of performance and computational efficiency. Being an architecture suitable for the transfer learning approach and having low computational complexity are two important parameters for the selection of the architecture. As it is known, the mobile systems on which the study focuses are devices with lower processor performance. For this reason, the low computational complexity of the architecture is an important advantage. The computational complexity of the MobileNet architecture also appears to be low, but its success in architectural classification accuracy is not sufficient compared to its competitors. In the study, it was decided to use EfficientNet architectures by evaluating their ability to achieve high success at low computational complexity, transfer learning ability and general applicability. The EfficientNet transfer learning method was employed for the classification of radiological chest X-ray pictures with deep learning models. The implementation of transfer learning facilitates greater success with reduced training data requirements (*Cinar & Kaya, 2022*). Moreover, the duration allocated to the educational part of architecture is diminishing. Figure 3 presents the block diagram of the EfficientNet-B0 architecture. The paramount component of the EfficientNet architecture is the MBConv blocks. 14-class classification output is obtained by giving radiological chest X-ray images as input data to the deep learning architecture. Instead of sigmoid activation function in the output layer, 14 class softmax activation function is used (*Aygun, Kaya & Kaya, 2022*).

Traditional CNN architectures scale in a single dimension to improve classification performance. This situation in previous CNN architectures causes parameter inefficiency and increases the computational cost. EfficientNet architecture, on the other hand, scales by optimizing the depth, width and resolution of the model using compound scaling. EfficientNet provides an optimal scaling between depth (d), width (w) and input image resolution (r) with the parameter $\varphi$ (phi). This scaling feature gives the architecture the

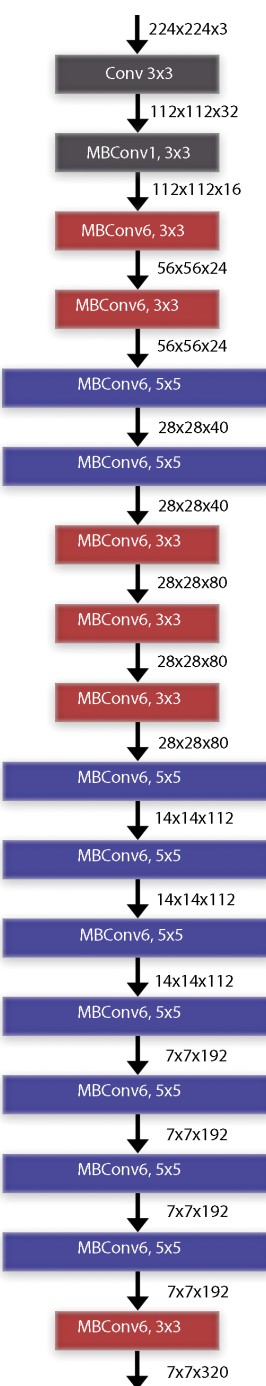

**Figure 3** EfficientNet deep learning architectures basic block diagram (EfficientNet-B0).

advantage of parameter efficiency. EfficientNet architectures are powered by the MBConv layer and the Swish activation function. The MBConv layer is an inverted extension of the traditional bottleneck layer with a predetermined expansion factor. After the expansion layer, each channel is processed in its own filter in the depthwise convolution layer. The

number of expanded channels is then reduced back to the channel size in the original image. The MBConv blocks used in the EfficientNet architecture include a squeeze-and-excitation attention (SE) module after each convolution process. The SE attention module is an important enhancement that can improve the classification performance of the architecture. Another important feature of the EfficientNet architecture is the use of the Swish activation function. The Swish activation function multiplies the input values by a sigmoid function and keeps the resulting negative values as small values instead of reducing them to zero. The fact that the negative values are not completely reduced to zero regulates the flow of gradients, resulting in a better optimization. This architectural improvement improves the learning process by providing a smoother gradient flow compared to the ReLU activation function used in many popular architectures.

## Coordinate attention mechanism

Coordinate attention mechanism is a mechanism developed to improve the efficiency of the feature extraction phase in convolutional neural networks (*Hou, Zhou & Feng, 2021*). While traditional attention mechanisms apply only channel-based or only spatial attention, the coordinate attention mechanism combines channel-based and spatial-based attention (*Niu, Zhong & Yu, 2021*). The architecture, which involves the addition of spatial information to channel attention, creates attention maps using two one-dimensional features to capture long-range dependencies and preserve spatial information, respectively.

Coordinate attention provides deep learning architectures with the ability to effectively process spatial information and improve feature extraction. In models using medical images as input, spatial and channel-based features can be extracted more accurately and contribute to increased diagnostic accuracy. When EfficientNet architecture and coordinate attention mechanism are used together, the goal of achieving higher classification success using fewer parameters is reinforced. It is fully compatible with the EfficientNet architecture by applying channel and spatial attention together in parallel. This combination can provide a hybrid architecture that is parameter efficient, fast and high-performing and can be used in the analysis of medical images.

The coordinate attention mechanism starts with average pooling, which preserves spatial information in the X and Y axes and provides global context per channel. Next, the features from the pooling layer are combined to obtain global context information and normalized using BatchNorm and non-linear. Following this process, feature maps are obtained by applying 2D convolution in parallel and separately on both axes (*Xu et al., 2024*). In the last step of the attention mechanism, the features are passed through a sigmoid activation function and then the feature maps are re-weighted to complete the attention mechanism phase. Figure 4 shows the block diagram of the coordinate attention mechanism.

## Proposed method

In this study, an autonomous disease diagnosis model is proposed with deep learning architectures using multiclass chest X-ray images containing data from 14 different diseases. For this purpose, EfficientNet architecture, a highly efficient and innovative convolutional neural network model, is used. In addition, the EfficientNet architecture

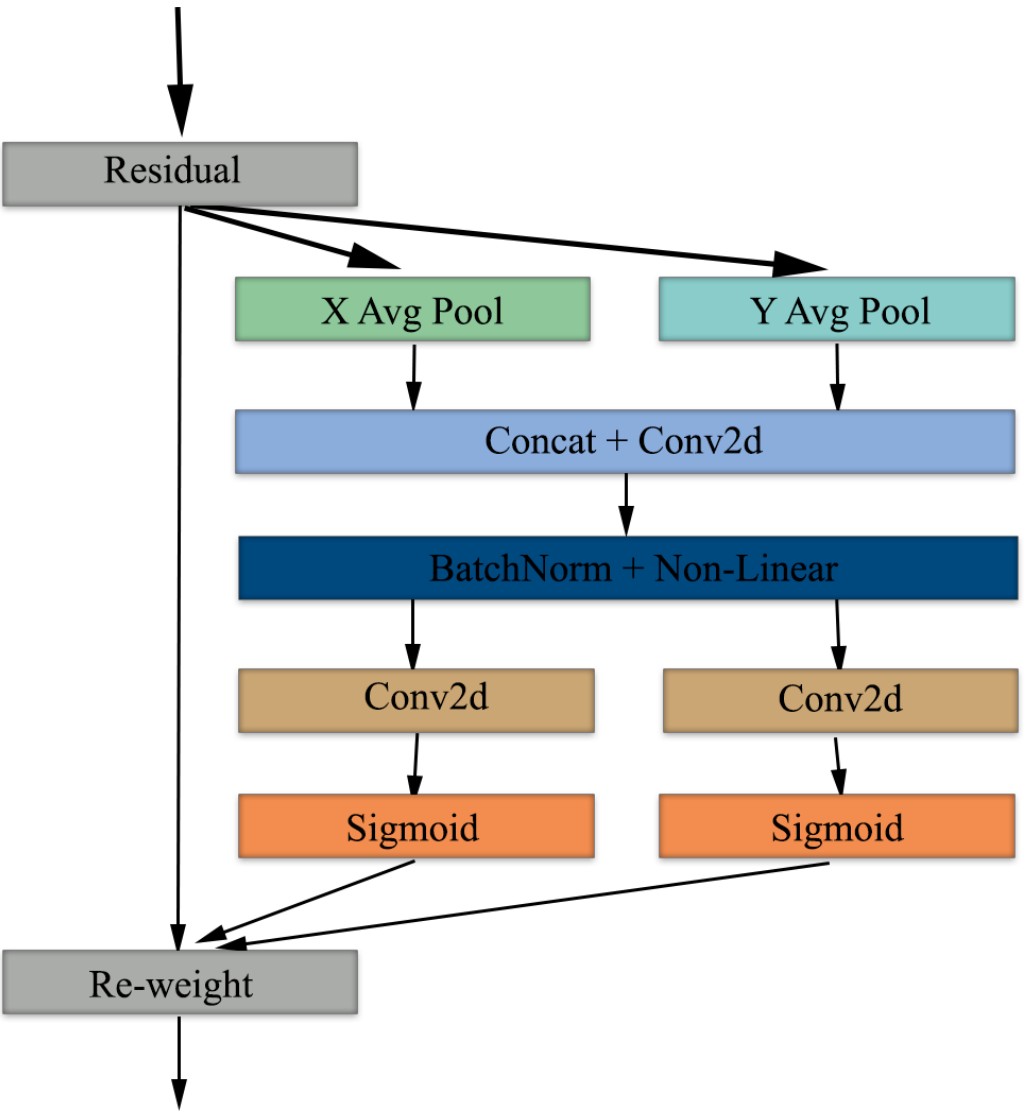

**Figure 4** Block diagram of the coordinate attention mechanism used in our study.

is improved using coordinate attention to contribute to the classification success of the EfficientNet architecture. EfficientNet was preferred in our study because it is an architecture that can skillfully capture even small details in medical images and achieves high success. In addition, the EfficientNet architecture is a deep learning architecture with high operational efficiency compared to other architectures. Another important reason for choosing EfficientNet architecture is that it is a competent architecture that can be used even in development environments with less GPU capacity.

The EfficientNet family of architectures scales in depth, width and resolution to create a low-complexity and high-performance convolutional neural network. While in popular traditional deep learning architectures the parameters that make up the architecture are determined manually, EfficientNet architectures use a systematic scaling method called

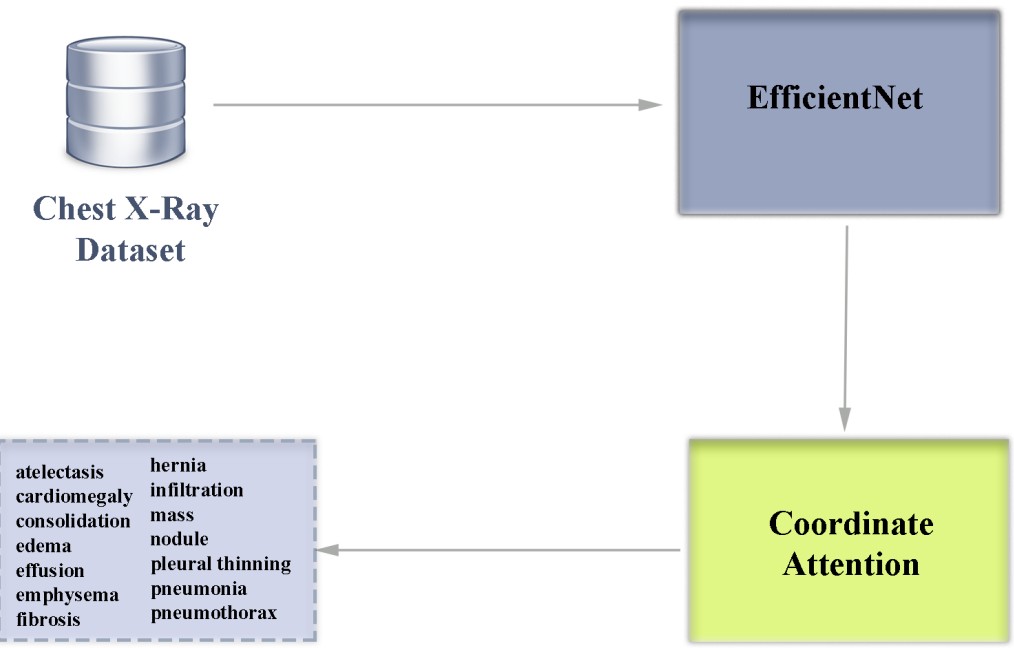

**Figure 5** Block diagram overview of our deep learning-based multiclass classification architecture enhanced with attention module.

compound scaling, which has the advantage of increasing the size of the model in a balanced way. In the EfficientNet family of architectures, there are scaling differences between models B0-B7. The B7 model is the EfficientNet version with the highest depth, width and resolution. Having a deeper network structure makes this architecture advantageous in capturing more detailed features in the analysis of diseases that can be detected by chest X-ray. The block diagram of our multi-class classification deep learning architecture used in our study is shown in Fig. 5.

Coordinate attention allows to increase the success of the feature extraction phase from visual inputs in solving problems such as disease diagnosis from medical images. When EfficientNet architecture and coordinate attention are used together, important features can be emphasized and the impact of irrelevant features can be reduced. This improvement preserves the lightweight structure of the EfficientNet architecture and increases the classification success. Within the scope of the study, a total of eight different architectures of the EfficientNet architecture between B0-B7 were examined and the results were analyzed in detail. In addition, coordinate attention was added to the model of EfficientNet architecture with the highest classification success and it was observed whether there was an increase in the success of the architecture.

The original labels in the dataset were used for the training, validation and testing processes. Afterwards, the system was tested with data that was not previously used in the training and validation phases. The proposed work has a framework that automates the disease diagnosis process from chest X-ray images and can support doctors in

**Table 2  Parameters used in the training phase of the proposed model.**

| Hyperparameter | Value |
| --- | --- |
| Optimizer | Adam |
| Metrics | Accuracy |
| Epochs | 20 |
| Initial learning rate | 1e−3 |
| Minimum learning rate | 1e−8 |
| Weight | ImageNet |
| Batch size | 32 |

making decisions during the diagnosis of diseases. This is a lightweight, fast and high-achieving approach that can support healthcare professionals to avoid wrong or incomplete treatments.

### Data and code availability

The ChestX-ray14 dataset, which is open source licensed and accepted as a benchmark dataset, was used in the training, validation and testing processes of the study. ChestX-ray14 is stored in a web archive that can be easily downloaded by all researchers who want to work on it (*National Institutes of Health Clinical Center, 2024*). The architecture codes used in the study are also available as supplemental files in the web version of the study. The images in the ChestX-ray14 dataset were subjected to some pre-processing steps in preparation for being given as input to the EfficientNet architecture. The dataset has a multi-folder structure in its original version, and in order to save it from this structure and turn it into a data stack, it was first collected in a single folder. In the next step, all images in the dataset were resized to 224 × 224, the input dimensions of the EfficientNet architecture.

The ChestX-ray14 dataset contains a total of 112,104 labeled images. Chest X-ray images are first split into 80% train_valid and 20% test. In the next step, the train_valid group is split into train and validation groups with 80% and 20% respectively. In the final case, there are 70,650, 17,663, 22,079 chest X-ray images in the training, validation and test groups, respectively. In the training phase of all experiments, the initial learning rate was 1e−3 and the Adam optimizer was used. Adam Optimizer performs better optimization of the network by reducing the number of parameters and shortening the training time. The training procedure was performed for a total of twenty epochs and thirty-two batch sizes. Table 2 contains a list of the hyperparameters used for a total of 8 EfficientNet models in the experiments.

In order to ensure the best performance of the deep learning model used in our study, the hyperparameter settings were carefully tuned and the same parameters were carefully used in all experiments. Our study focuses on an efficient and fast learning process, therefore, man-optimization is preferred in the study, taking into account momentum and adaptive learning rate properties. An accuracy evaluation metric was chosen to ensure parameter optimization at training time. The learning rate parameter, which is directly related to the stability of the model during training, was initially set as 1e−3. In order to optimize the learning process and manage the optimization process in small steps, it was reduced to

1e−8. Training the architecture for 20 epochs was determined by considering overlearning and having an efficient learning process. The batch size of 32 was chosen as a trade-off based on the GPU memory utilization and training time for which the architecture was trained. All models were initialized with ImageNet weights, thus speeding up the training process and taking advantage of pre-trained features. Since the dataset used in the study is a very large set with 112,120 images and considering the computational burden of retraining the dataset by dividing it into folds, the model's generalization ability was tested by dividing the dataset into training, validation and test subsets instead of the cross-validation strategy. In addition, it was observed that other studies in the literature generally use fixed training, validation and test subsets, and a similar strategy was followed for consistency with the literature.

### Computing infrastructure

Google Colab platform was used for training, validation and testing of the proposed method. Google Colab offers a free and paid version of the platform where we can run code for machine learning and deep learning architectures. The paid versions of the platform offer higher RAM and GPU capacity. The high number of images in the dataset we used in our study required high RAM and computational power, so Colab pro+, the paid version, was preferred in our research. With this advantage, a high-performance NVIDIA Tesla T4 GPU was used in our research and a fast and efficient experimental environment was used with the advantage provided by the hardware. In addition, due to the large number of images and the high size of the data set, the experiments were carried out using the 52 GB RAM option.

## RESULTS

Many experiments have been carried out to test the effectiveness of the proposed deep learning architecture. Labeled images in the data set may belong to more than one disease. For this reason, area under the curve (AUC) data was used to compare the effectiveness of the proposed model with other architectures in the literature (*Korkmaz & Boyacı, 2021*; *Song et al., 2022*).

AUC, a popular classification model evaluation metric, is used to create an optimized learning model and compare the performance of different learning models (*Fırat et al., 2022*). Unlike probability measures, the AUC value reflects the overall ranking performance of a classifier. The AUC value for the two-class problem is calculated as follows (*Hand & Till, 2001*).

$$AUC = \frac{S_p - n_p(n_n + 1)/2}{n_p n_n}. \tag{1}$$

In Eq. (1), $S_p$ is the sum of all positive samples. $n_p$ and $n_n$ indicate the number of positive and negative samples, respectively. It has been proven by studies in the literature that the AUC value is theoretically and empirically better than the accuracy metric in order to evaluate the performance of classification training success and to distinguish the optimal solution (*Huang & Ling, 2005*).

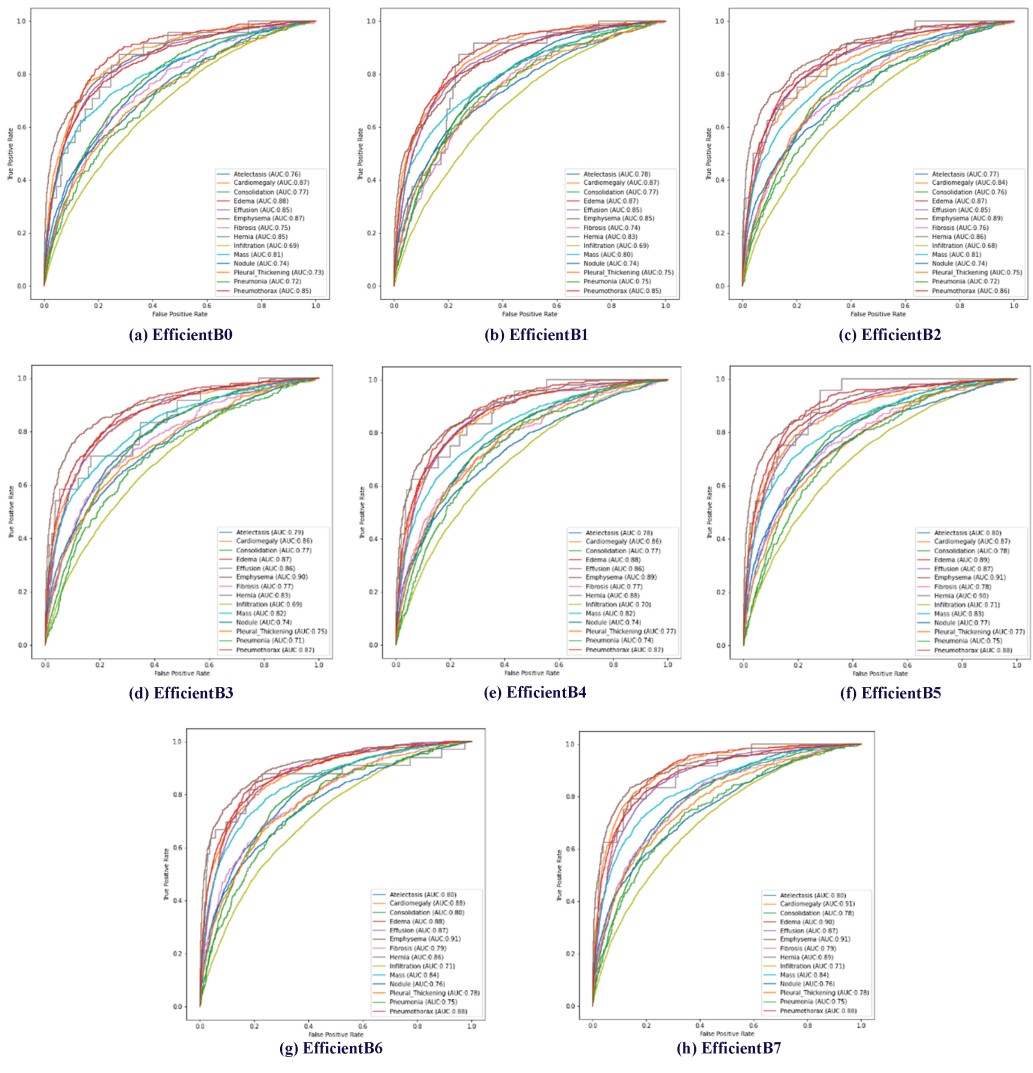

**Figure 6  Multiclass receiver operating characteristics (ROC) of the EfficientNetB0-B7 models.**

The effectiveness of the proposed method has been tested using AUC data. Receiver operating characteristic (ROC) curves of 14 diseases in the data set are shown in Fig. 6. In addition, the calculated AUC values for each class are given in Fig. 6. In addition, the calculated AUC values for all classes are given in the lower right corner of each of the ROC graphs shown for all models trained in Fig. 6.

In EfficientNet models, the larger the model number, the larger the parameter number of the model. The EfficientNetB7 model uses approximately 11 times more parameters than the EfficientNetB0 model. Increasing the number of parameters used in EfficientNet models is expected to increase the model's processing time and success rate. In addition, it is expected that the model depth will directly affect the success rate. Within the scope of determining the proposed method, a total of eight models between EfficientNet B0 and B7 were trained using the same parameters and classification results were obtained based on

**Table 3  AUC (Area Under The curve) results values of EfficientNetB0-B7 models on ChestX-ray14.**

| Pathology | EfficientNetB0 | EfficientNetB1 | EfficientNetB2 | EfficientNetB3 | EfficientNetB4 | EfficientNetB5 | EfficientNetB6 | EfficientNetB7 |
|---|---|---|---|---|---|---|---|---|
| Atelectasis | 0.76 | 0.78 | 0.77 | 0.79 | 0.78 | 0.80 | 0.80 | 0.80 |
| Cardiomegaly | 0.87 | 0.87 | 0.84 | 0.86 | 0.86 | 0.87 | 0.88 | 0.91 |
| Consolidation | 0.77 | 0.77 | 0.76 | 0.77 | 0.77 | 0.78 | 0.80 | 0.78 |
| Edema | 0.88 | 0.87 | 0.87 | 0.87 | 0.88 | 0.89 | 0.88 | 0.90 |
| Effusion | 0.85 | 0.85 | 0.85 | 0.86 | 0.86 | 0.87 | 0.87 | 0.87 |
| Emphysema | 0.87 | 0.85 | 0.89 | 0.90 | 0.89 | 0.91 | 0.91 | 0.91 |
| Fibrosis | 0.75 | 0.74 | 0.76 | 0.77 | 0.77 | 0.78 | 0.79 | 0.79 |
| Hernia | 0.85 | 0.83 | 0.86 | 0.83 | 0.88 | 0.90 | 0.86 | 0.89 |
| Infiltration | 0.69 | 0.69 | 0.68 | 0.69 | 0.70 | 0.71 | 0.71 | 0.71 |
| Mass | 0.81 | 0.80 | 0.81 | 0.82 | 0.82 | 0.83 | 0.84 | 0.84 |
| Nodule | 0.74 | 0.74 | 0.74 | 0.74 | 0.74 | 0.77 | 0.76 | 0.76 |
| Pleural thickening | 0.73 | 0.75 | 0.75 | 0.75 | 0.77 | 0.77 | 0.78 | 0.78 |
| Pneumonia | 0.72 | 0.75 | 0.72 | 0.71 | 0.74 | 0.75 | 0.75 | 0.75 |
| Pneumothorax | 0.85 | 0.85 | 0.86 | 0.87 | 0.87 | 0.88 | 0.88 | 0.88 |
| Average AUC | 0.7957 | 0.7957 | 0.7971 | 0.8021 | 0.8093 | 0.8221 | 0.8221 | 0.8265 |

disease classes. Table 3 presents the test results of multiclass chest X-ray images on eight different EfficientNet models.

As expected, the EfficientNetB7 model showed the highest classification success on average, as it has a higher parameter number and depth than other EfficientNet models. The EfficientNetB7 model achieved an average classification success of 0.8265 AUC across 14 disease classes. When the chest X-ray disease classes were examined separately, the EfficientNetB7 model achieved the highest success rate in 11 of the 14 diseases. However, other models achieved the highest classification success in three diseases. In consolidation disease, the EfficientNetB6 model achieved a classification success value of 0.80 AUC. When we compare the EfficientNetB6 and EfficientNetB7 models for consolidation disease, there appears to be a difference of about 2%. In hernia disease, the highest classification success was achieved in the EfficientNetB5 model. In hernia disease, the EfficientNetB5 model achieved a classification success value of 0.90 AUC. When we compare the EfficientNetB5 and EfficientNetB7 models for hernia disease, there appears to be a difference of about 1%. Finally, the highest classification success in nodule disease was obtained in the EfficientNetB5 model. In nodule disease, the EfficientNetB5 model achieved a classification success value of 0.77 AUC. When we compare the EfficientNetB5 and EfficientNetB7 models for nodule disease, there appears to be a difference of about 1%.

Data sets used in machine learning and deep learning fields are generally divided into three groups as train, validation and test. In this study, the data set was divided into three subgroups using the ratios described in the previous sections. The validation subset is the part of the dataset used to evaluate the performance of the model during the training phase. In addition, the validation dataset provides a testing platform to determine which

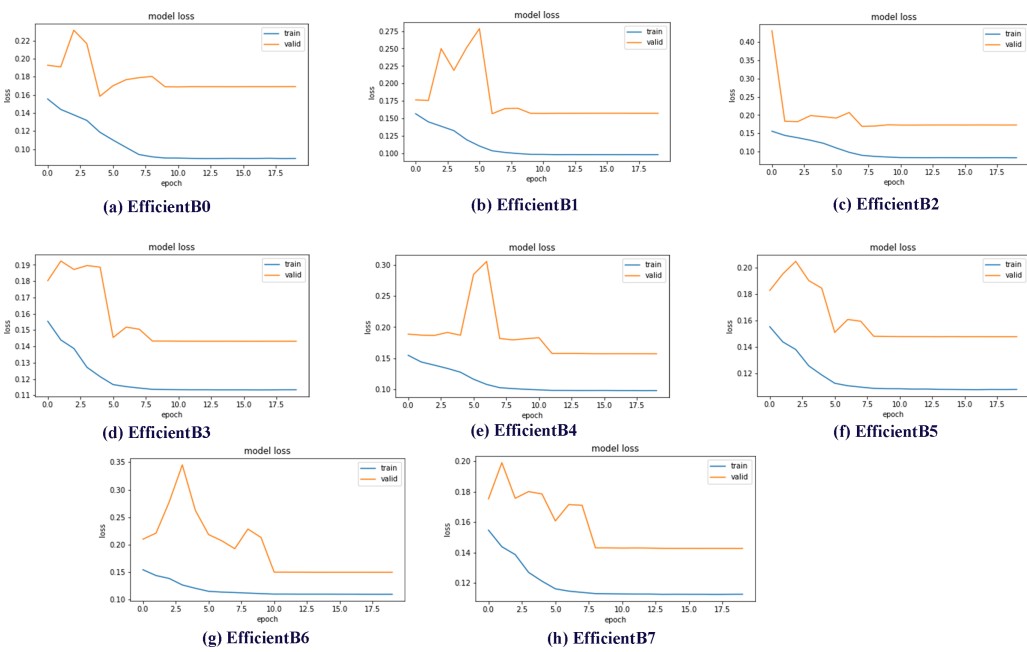

**Figure 7  EfficientNetB0-B7 models loss chart.**

model performs well and to set the optimal parameters for the models. Within the scope of the studies, eight different models, EfficientNetB0-B7, were trained and test results were obtained. One of the best demonstrations of comparing the performance of models in validation subsets is model loss charts. By examining the model loss graphics, it is clearly seen how well the model performs at which epoch during the training and testing phases. Model loss graphs of EficientNetB0-B7 models are shown in Fig. 7.

By examining the model loss graph, it can also be analyzed in which epoch the models start to perform best. When Figs. 7A graph is examined, it is seen that the EfficientB0 model draws a nearly linear graph after the 10th Epoch. This graph shows that the model did not gain a significant amount of learning in the trainings after the 10th Epoch. When the Figs. 7B graph is examined, it is seen that the EfficientB1 model draws a nearly linear graph after the 8th Epoch. When the graph in Figs. 7E is examined, it is seen that the EfficientB4 model draws a nearly linear graph after the 11th Epoch. One of the main reasons why the graph approaches linear at different epochs is that the models have a different number of layers.

Classification achievements, namely AUC data, were calculated for all classes. In Table 4, the proposed method for all classes of the ChestX-ray14 data set and the results of other recent studies in the literature are shown. The average classification success AUC of the proposed method was calculated as 0.8265. This value is the highest classification success compared to other publications in the compared literature. When the classes are examined separately; The highest classification success in atelectasis, cardiomegaly, edema, effusion, infiltration, mass, pneumonia and pneumothorax classes was obtained with the proposed method. There are 14 different diseases in the data set used as described in the previous

**Table 4  Quantitative comparison of the best result proposed in our study with other similar studies in the literature using the AUC evaluation metric.** The highest value for each class is labeled with a bold color so that the highest result in disease classes and mean values can be seen more clearly.

| Pathology | Huang & Ling (2005) | Gündel et al. (2019) | Guan & Huang (2020) | Wang et al. (2017) | Yao et al. (2017) | (Our) EfficientNet without attention | (Our) EfficientNet enhanced with coordinate attention |
|---|---|---|---|---|---|---|---|
| Atelectasis | 0.7720 | 0.7670 | 0.7810 | 0.7790 | 0.7850 | 0.8020 | **0.8130** |
| Cardiomegaly | 0.9040 | 0.8830 | 0.8800 | 0.8950 | 0.8830 | **0.9070** | 0.8950 |
| Consolidation | 0.7880 | 0.7450 | 0.7540 | 0.7590 | 0.7810 | 0.7830 | **0.7880** |
| Edema | 0.8820 | 0.8350 | 0.8500 | 0.8550 | 0.8880 | **0.8990** | 0.8970 |
| Effusion | 0.8590 | 0.8280 | 0.8290 | 0.8360 | 0.8690 | 0.8740 | **0.8780** |
| Emphysema | 0.8290 | 0.8950 | 0.9080 | **0.9330** | 0.8790 | 0.9060 | 0.9190 |
| Fibrosis | 0.7670 | 0.8180 | 0.8300 | **0.8380** | 0.7490 | 0.7940 | 0.7920 |
| Hernia | 0.9140 | 0.8960 | 0.9170 | **0.9380** | 0.8060 | 0.8950 | 0.8970 |
| Infiltration | 0.6950 | 0.7090 | 0.7020 | 0.7100 | 0.6960 | 0.7110 | **0.7160** |
| Mass | 0.7920 | 0.8210 | 0.8340 | 0.8340 | 0.8120 | 0.8360 | **0.8470** |
| Nodule | 0.7170 | 0.7580 | 0.7730 | **0.7770** | 0.7140 | 0.7560 | 0.7710 |
| Pleural thickening | 0.7650 | 0.7610 | 0.7780 | **0.7910** | 0.7580 | 0.7770 | 0.7750 |
| Pneumonia | 0.7130 | 0.7310 | 0.7290 | 0.7370 | 0.7150 | 0.7470 | **0.7620** |
| Pneumothorax | 0.8410 | 0.8460 | 0.8570 | 0.8780 | 0.8620 | **0.8840** | 0.8820 |
| Average AUC | 0.8027 | 0.8066 | 0.8159 | 0.8257 | 0.7997 | **0.8265** | **0.8309** |

sections. The test results clearly showed that the proposed method achieved the most successful results in eight of the 14 disease classes. The highest value for each class is labeled with a bold color so that the highest result in disease classes and mean values can be seen more clearly.

The highest classification success in atelectasis disease was obtained using the proposed method with a success value of 0.8020 AUC. This rate obtained in atelectasis disease is approximately 2.1% higher than its closest competitor. The highest success rate in cardiomegaly disease was obtained with the recommended method with a success rate of 0.9070 AUC. The results of other studies examined in the literature also obtained similar success rates. The highest success rate in consolidation disease was obtained in the study (*Yao et al., 2017*), which was reviewed in the literature. However, there is only 0.5% difference between the result obtained within the scope of the proposed study and the highest result. The highest success in Edema disease was obtained by using the proposed method. The proposed method achieved a success of 0.8990 AUC. Compared to the competitors examined in the literature, 1.7% higher success was achieved than the closest competitor. It achieved the highest success with a value of 0.8740 AUC using the recommended method in effusion disease. 1.5% better success than the closest competitor was obtained with the proposed method. The highest success rate in Emphysema disease was obtained in the study numbered (*Wang et al., 2021*), which was reviewed in the literature. However, there is a 2.7% difference between the result obtained within the scope of the proposed study and the highest result. The highest success rate in fibrosis

disease was obtained in the study numbered (*Wang et al., 2021*), which was reviewed in the literature. There is a 4.4% difference between the result obtained within the scope of the proposed study and the highest result. The highest success rate in hernia disease was obtained in the study numbered (*Wang et al., 2021*), which was reviewed in the literature. There is a 4.3% difference between the result obtained within the scope of the proposed study and the highest result. The proposed method in infiltration disease has achieved the highest rate of success with a small difference. The proposed method achieved an AUC of 0.7110 for infiltration disease. It achieved a success rate of 0.1% higher than the closest competitor examined in the literature. In mass disease, the proposed method achieved the highest success with a small difference. The proposed method achieved an AUC of 0.8360 for mass disease. The success rate was 0.2% higher than the closest competitor examined in the literature. The highest success rate in nodule disease was obtained in the study numbered (*Wang et al., 2021*), which was reviewed in the literature. There is a 2.1% difference between the result obtained within the scope of the proposed study and the highest result. The highest success rate in pleural thickening disease was obtained in the study (*Wang et al., 2021*), which was reviewed in the literature. There is a 1.4% difference between the result obtained within the scope of the proposed study and the highest result. The highest classification success in pneumonia was obtained using the recommended method with a success value of 0.7470 AUC. This rate obtained in pneumonia is a success rate of approximately 1.0% higher than its closest competitor. Finally, the proposed method in pneumothorax disease achieved the highest success with a small difference. The proposed method achieved an AUC of 0.8840 for pneumothorax disease. The success rate was 0.6% higher than the closest competitor examined in the literature.

The accuracy evaluation metric is widely used by researchers to measure overall model performance in classification problems. In deep learning architectures, it is a numerical measure of how well the model's predicted labels match the actual labels generated by the doctors and is obtained by dividing the total number of correct predictions by the total number of test samples. Our study focuses on the multiclassification problem with the pulmonology dataset. This shows that multiple diseases can be detected from an X-ray image. As in real world problems, the case where a patient can have multiple diseases at the same time is labeled in the dataset. For this reason, sigmoid was preferred as the activation function instead of softmax as in classical multiclass classification problems. The sigmoid function generates probability values for each disease. When calculating the accuracy evaluation metric, a threshold value of 0.5 was used for this probability. The choice of this threshold value means that if the prediction probability is 50% and above, the disease is considered to be present and if it is below, the disease is considered to be absent. Thanks to this method, the accuracy evaluation metric could be calculated for the multiclass classification problem. In Table 5, the results of the accuracy evaluation metric calculated for the EfficientNet architecture in its raw form and enhanced with coordinate attention are shared on the basis of disease class.

When the results shown in Table 5 are analyzed, it is seen that the coordinate attention mechanism provides a significant accuracy improvement on the EfficientNet architecture. Especially in fibrosis, pneumonia and emphysema diseases, the use of the

**Table 5** The effect of coordinate attention on accuracy evaluation metric in EfficientNet architecture for chest X-ray pathology classification.

| Pathology | EfficientNet without attention | EfficientNet enhanced with coordinate attention |
|---|---|---|
| Atelectasis | 89.61 | 90.43 |
| Cardiomegaly | 97.63 | 98.34 |
| Consolidation | 96.18 | 96.95 |
| Edema | 98.08 | 98.86 |
| Effusion | 89.58 | 90.45 |
| Emphysema | 98.11 | 98.90 |
| Fibrosis | 98.58 | 99.40 |
| Hernia | 99.89 | 99.89 |
| Infiltration | 82.24 | 82.92 |
| Mass | 95.15 | 96.02 |
| Nodule | 94.41 | 95.23 |
| Pleural thickening | 97.40 | 98.20 |
| Pneumonia | 98.89 | 99.68 |
| Pneumothorax | 95.61 | 96.35 |
| Average accuracy | 95.09 | 95.83 |

attention mechanism has a significant effect on classification success. In the average of 14 diseases in the dataset, the accuracy rate increased from 95.09% to 95.83% with the addition of the attention mechanism. This increase in classification success shows that the coordinate attention mechanism, which combines spatial attention and channel attention techniques, supports the feature extraction phase. The results of the experiments show that an EfficientNet model enhanced with coordinate attention can provide more reliable diagnostic support mechanisms in clinical applications and support doctors in making decisions during the diagnosis of diseases.

## DISCUSSION

When a radiologist or medical doctor who is an expert in his field examines the radiology image of a patient, he does not always encounter lesions belonging to only one disease. Because patients can sometimes have more than one disease at the same time. Some of the radiology images in the ChestX-ray14 dataset have only one disease, but some have findings from more than one disease. Figure 8, shows 12 randomly selected test images. On the chest X-ray images, the estimation values of which class they belong to are given as a percentage.

In the upper part of each chest X-ray image given in the figure, the actual classes and predictive classes to which the picture belongs are given, respectively. The first chest X-ray image with index number 25 in Fig. 8 is labelled with Infiltration disease in the csv file provided in the dataset. In the estimations made using the proposed method, it was correctly estimated as Infiltration disease for the chest X-ray image with index number 25. Although a multi-label prediction model has been developed, the developed model gave

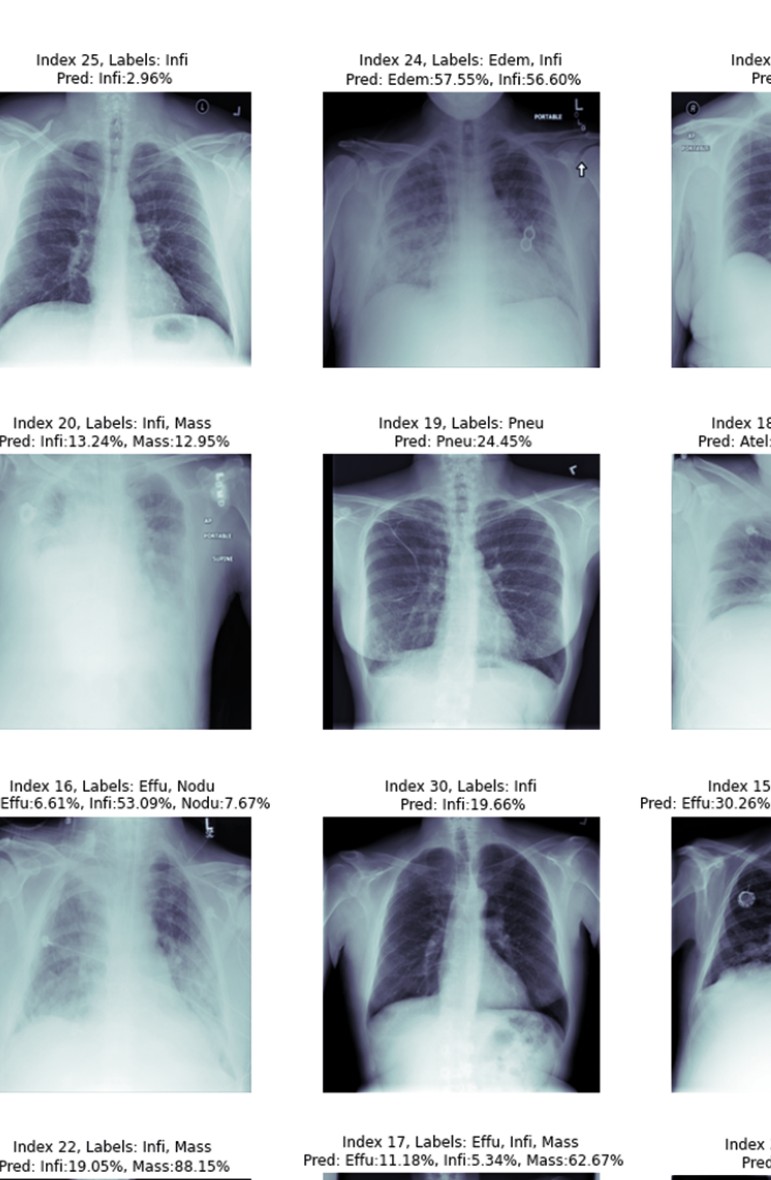

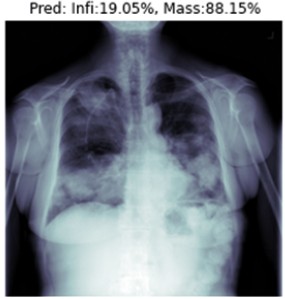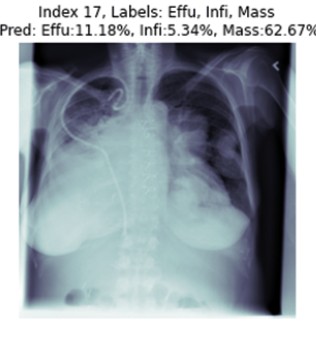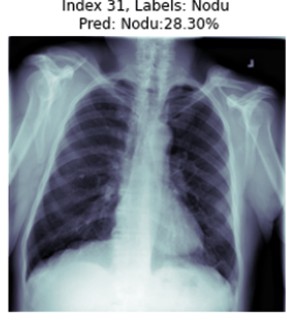

**Figure 8** **Proposed method image prediction examples.**

accurate results for only 1 disease. This shows that the model is working perfectly for the first image, but the probability given in this image is slim.

The chest X-ray image with index number 24 given in the second row in Fig. 8 is labelled with edema and infiltration diseases in the csv file given in the dataset. We can explain this situation more clearly as follows: At the time of the chest X-ray image, the patient has symptoms of two diseases at the same time. In the estimation of the proposed method, the findings of edema and Infiltration diseases were found to be correct in the chest X-ray image with index number 24. The proposed deep learning model reports that the test image with index number 24 contains findings of Infiltration disease with 57.55% similarity to Edema and 56.60% similarity rate. Chest X-ray image with index number 17 in the eighth row in Fig. 8 is labelled with effusion, infiltration and mass diseases in the csv file given in the dataset. This situation showed that; At the time of the chest X-ray image, the patient has symptoms of three diseases at the same time. In the estimation of the proposed method, the findings of effusion, infiltration and mass diseases were found to be correct in the chest X-ray image with index number 17. The proposed deep learning model reports that the test image with index number 24 contains findings of Effusion with 11.18% similarity, Infiltration with 5.34% similarity, and mass disease with 62.67% similarity. The test results of the chest X-ray image with index number 17 showed that the proposed method can correctly classify a chest X-ray image containing findings from three diseases.

Chest X-ray image with index number 16 in the ninth row in Fig. 8 is labelled with effusion and nodule diseases in the csv file given in the dataset. In the estimation made by the proposed method, it was estimated that there were findings of effusion, infiltration and nodule diseases in the chest X-ray image with index number 16. The proposed deep learning model reports that the 16 index test image contains findings of effusion with 6.61% similarity, Infiltration with 53.09% similarity, and nodule disease with 7.67% similarity. The test results of the chest X-ray image with index number 16 showed that the proposed method made the classification by estimating the details incorrectly. In fact, there is no problem in detecting the diseases existing in the 16 index image. The proposed method correctly predicted that the patient had effusion and nodule diseases. However, the findings of a disease that was not labelled in the data set were additionally detected by the proposed method. Since the deep learning architecture detects the Infiltration disease with a very strong probability, it is thought that this problem can be eliminated in further studies by examining the data set by experts in the field or using different deep learning architectures.

In order to test the effectiveness of the proposed model, analyses were made based on disease classes. The mean success value is expressed by AUC data. It is important that all classes are estimated at higher achievements so that the mean can be high. In addition, the poor performance of the model in certain diseases may lead to incorrect treatment of patients during real-life use of the application. For these purposes, the tested models were plotted for all classes and the worst predicted classes were analysed visually. Figure 9 shows the comparison of AUC values based on disease classes graphically.

As can be clearly seen from Fig. 9, Infiltration disease was predicted with the lowest success in all models. Infiltration disease was best predicted in the EfficientNetB7 model with an AUC of 0.71. Infiltration disease was predicted in the worst EfficientNetB2 model

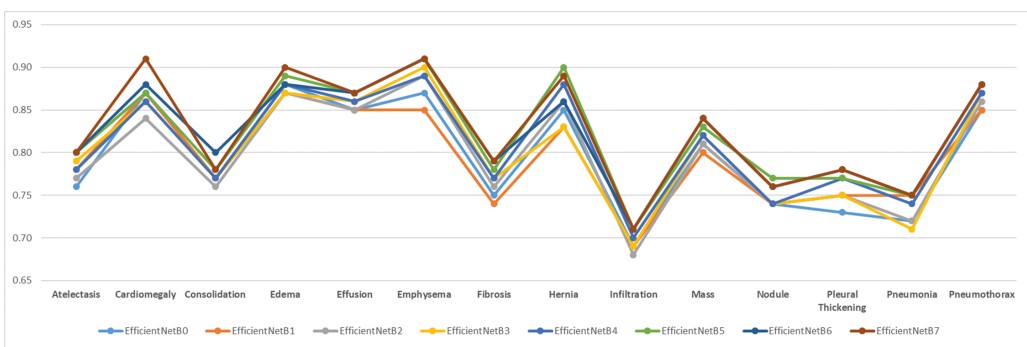

**Figure 9** Comparison of AUC performance values according to data set disease classes.

with a value of 0.68 AUC. The average predictive success of infiltration disease in the 8 tested models was 0.70 AUC. It was clearly seen that the same disease was predicted poorly with a similar rate in other models in the literature. The overall success of the study is also declining due to poor prediction of infiltration disease. For this reason, in future studies, it is considered to apply data augmentation algorithms specifically to the relevant class or to find new examples belonging to the relevant class.

It should be noted at this point that there are 112,104 labelled images in the ChestX-ray14 dataset. By showing the data set to new radiologists or doctors who are experts in the field, it may be concluded that the estimation for index 14 is more accurate. In addition, the fact that it provides accurate predictions for images belonging to more than one class, at least in some of the diseases, constitutes a serious source of motivation for further studies.

## Limitations

The results show that the deep learning-based classification model proposed in this study achieves high success in disease detection from chest X-ray images. However, the experiments have some limitations such as data dependency and real-time usage. Images from the ChestX-ray14 dataset were used in all experiments. The dataset used is a comprehensive dataset that is considered a benchmark and contains a large number of images, and at the same time, it provided the opportunity to train, validate and test with a high number of images. However, the dataset consists only of chest radiographs and does not contain images related to diseases of other parts of the human body. The limitations of this study are the use of the images in the ChestX-ray14 dataset and focusing only on chest diseases. The test data used in the study was obtained by dividing the dataset into training, validation and test subsets. No integration and real-time testing with medical imaging devices in real-world hospitals was performed at this stage. The results obtained in the model proposed in the study will be able to support expert radiologists in making decisions in the diagnosis of diseases. In the diagnosis scenario designed with the support of autonomous systems, the diagnosis of diseases can be accelerated and the accuracy rate can increase. In addition, the applications using the proposed method have the potential to be an important resource for medical students receiving specialty training.

The other limitations of our study are generalizability, data reliability and ethical considerations, which are important cornerstones in drawing the boundaries of the study. The generalizability of the model is directly related to the data set used. The dataset used in the study is one of the datasets with the highest number of images in the field, which is considered a benchmark in the literature. The fact that the data is obtained from many patients and used by many researchers and that it consists of data from real hospitals reveals the success of the dataset in terms of generalizability parameter. In terms of data reliability, the dataset was labeled by expert radiologists before it was shared as open access. Afterwards, these data were used in many studies and many studies were conducted on them. No reports of incorrect image label pairs in the open-access dataset were found in the studies, but this constitutes the limitations of the study. In terms of ethical aspects of the study, the dataset is open to the public and has been used with the necessary permissions, there is no private information such as patients' identity information. It is planned that the results obtained and the architecture developed can support expert doctors in making decisions in the diagnosis of diseases. In order to prevent possible erroneous predictions and misdiagnoses, using it in a way to work with clinical experts in decision processes and supporting it with explainability methods will reduce ethical problems and increase patient safety.

## CONCLUSIONS

In this study, chest X-ray images are trained using EfficientNet convolutional neural network architectures and coordinate attention mechanism. Training, verification and testing processes were carried out with the same parameters in eight different models in total. Classification success of chest X-ray images in the dataset is given by ROC curves, AUC values, and class-based overall success rates. Training, verification and testing were carried out in nine different architectures, taking into account the computational complexity and classification performance. In the analyses made at the end of the testing process, the EfficientNet-B7 architecture had the highest success in solving the 14-class classification problem, obtaining an AUC value of 0.8265. The EfficientNet enhanced with coordinate attention architecture achieved a classification success with an AUC value of 0.8309. As a result, the findings obtained within the scope of the study show that disease classification can be made successfully from chest X-ray images using deep learning architectures. The findings show that the proposed approach can create a decision-support mechanism for specialist doctors in the detection of diseases. The fact that the proposed architecture can achieve high classification success despite having fewer parameters encourages researchers to use the study in real-life applications. Thanks to a faster architecture with less computational complexity, deep learning-based disease diagnosis opportunities can be used in hospitals with less infrastructure in rural areas. In addition, the use of the study in the diagnosis of diseases by doctors who are experts in their field will make significant contributions to providing more accurate and faster treatment.

The limitations of this study were determined by the ChestX-ray14 dataset and 14 different diseases in it. Other diseases that can be detected with chest X-ray images were not

evaluated within the scope of the study. Although the study can achieve significant results in the experimental environment, it has not been tested with real-time data in the clinical environment. Working with real-time data requires ethical permissions and diplomatic processes, and studies on the real-time use of the research in clinics are ongoing. In future studies, it is aimed to improve the classification success of the model by developing new deep learning architectures specific to chest X-ray images, integrating different attention mechanisms and using metaheuristics (*Kaya, Kaya & Alhajj, 2016*) in the feature extraction phase. It is planned to expand the dataset to eliminate heterogeneous data distribution and to better represent rare diseases. It is also planned to test the developed deep learning architecture in real-time clinical environments to prepare the ground for its use in hospitals and to optimize it for low-equipped devices. These studies reveal the potential to shorten the runtime and increase the success of deep learning-based autonomous disease diagnosis systems.

### Funding
This study was supported by the Scientific and Technological Research Council of Turkey (TUBITAK) under Grant No: 123E171. The funders had no role in study design, data collection and analysis, decision to publish, or preparation of the manuscript.

### Grant Disclosures
The following grant information was disclosed by the authors:
Scientific and Technological Research Council of Turkey (TUBITAK): No: 123E171.

### Competing Interests
The authors declare there are no competing interests.

### Author Contributions
- Murat Ucan conceived and designed the experiments, performed the experiments, analyzed the data, performed the computation work, prepared figures and/or tables, and approved the final draft.
- Buket Kaya conceived and designed the experiments, performed the experiments, performed the computation work, prepared figures and/or tables, and approved the final draft.
- Osman Aygun analyzed the data, authored or reviewed drafts of the article, and approved the final draft.
- Mehmet Kaya conceived and designed the experiments, performed the experiments, analyzed the data, authored or reviewed drafts of the article, and approved the final draft.
- Reda Alhajj analyzed the data, authored or reviewed drafts of the article, and approved the final draft.

## Data Availability

The National Institutes of Health Chest X-Ray dataset is available at: https://nihcc.app. box.com/v/ChestXray-NIHCC, and https://www.kaggle.com/nih-chest-xrays/data.

## Supplemental Information

Supplemental information for this article can be found online at http://dx.doi.org/10.7717/ peerj-cs.2968#supplemental-information.

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
