# Peer review of "Comparison of EfficientNet CNN models for multi-label chest X-ray disease diagnosis"

_PeerJ Computer Science, doi:10.7717/peerj-cs.2968_

## Round 0.1 · original submission · Major Revisions

· Academic Editor

Major Revisions

The reviewers have substantial concerns about this manuscript. The authors should provide point-to-point responses to address all the concerns and provide a revised manuscript with the revised parts being marked in different color.

Reviewer 2 ·

Basic reporting

• The English language needs to be improved, and it is recommended that paragraphs be paraphrased to achieve more coherent and cohesive sentences for better understanding.
• In the introduction section, paragraphs 1 and 2 authors presented information without any references. Therefore, it is recommended to add reference citations.
• It is better to start the introduction without this sentence, “Thanks to the developing medical imaging techniques." It is not recommended in academic writing.
• The introduction should be well organised, focusing on the main keywords mentioned in the manuscript title.
• It is recommended to move the reviewed works to a separate section (2 literature review) and review more articles.

Experimental design

• The authors used the EfficientNet family, which consists of 8 predefined CNN-based models (B0 to B7).
• There is no modification or enhancement for the model architecture.
• The authors did not present clarification about the architecture of the EffecientNet model.

Validity of the findings

The results are measured only in terms of the AUC metric. There are other evaluation metrics such as accuracy, F1, and recall that can be used to indicate a better understanding of the results.

Annotated reviews are not available for download in order to protect the identity of reviewers who chose to remain anonymous.

·

Basic reporting

The manuscript, titled 'Comparison of EfficientNet CNN models for multi- label chest X-Ray disease diagnosis' presents significant technical contributions to the field. However, its clarity, accessibility, quality and impact could be further improved by addressing the following points:

1. Abstract: Clearly state the novelty (EfficientNetB7 for multi-label chest X-ray classification), quantify performance gains (e.g., AUC improvement), and briefly mention real-world applications (e.g., aiding radiologists).
2. Introduction: Define the research gap by detailing the limitations of existing methods (complexity, accuracy, real-world use). Justify the EfficientNet choice (efficiency/accuracy balance).

Experimental design

3. Dataset: Address class imbalance (specify the method). Justify image resizing and split ratios, explaining their impact.
4. Model Architecture: Simplify the EfficientNet description (consider a visual diagram). Justify using EfficientNetB7 (superior multi-label performance).
5. Experimental Design: Describe hyperparameter tuning and the rationale behind chosen values. State whether cross-validation was used.

Validity of the findings

6. Results: Include qualitative analysis (visual examples). Analyze why EfficientNetB7 performs better (multi-label handling, complex cases).
7. Discussion: Discuss limitations (data reliance, unseen datasets, real-time use). Expand on clinical applications (diagnosis assistance).
8. Figures/Tables: Integrate figures (ROC curves, loss charts) better into the text. Improve table visual appeal.
9. Conclusion: Suggest specific future work (other architectures, real-time testing). Emphasize broader implications (improved healthcare).

Additional comments

10. Limitations: Expand this section to address generalizability, data reliance, and ethical considerations.
11. Language: Simplify complex sentences. Ensure clear, professional language. Consider professional editing. (Example: "The proposed method achieved an AUC of 0.8360 for Mass disease.")
12. Reproducibility: Include a "Data and Code Availability" section with links and detailed instructions. Describe preprocessing step-by-step.
13. References: Update with recent work (EfficientNet, multi-label classification) such as Detection of tuberculosis from chest X-ray images: Boosting the performance with vision transformer and transfer learning; Automatic detection of Covid-19 from chest X-ray and lung computed tomography images using deep neural networks and transfer learning

---

## Round 0.2 · Minor Revisions

· Academic Editor

Minor Revisions

There are some remaining minor concerns that need to be addressed.

Reviewer 2 ·

Basic reporting

1. The manuscript is well-structured with comprehensive details for each section.

2. There are some unnecessary repetitions of some phrases, which I have indicated in the attached PDF file.

3. In my first revision of the research, I recommended making a separate section for previous studies (literature review), and it cannot be a subsection in the introduction section, as this section is important in drawing out the gaps between previous studies.

4. The tables and figures are well-organised according to the journal standards.

Experimental design

1. It is important to focus on the preprocessing techniques that have been utilized to prepare data for training.

2. The time needed to train the model is a very important factor in your research contribution. In addition, refer to the hardware specifications (CPU, GPU, and RAM size) used to train and test the model.

Validity of the findings

No Comment.

Additional comments

No comment.

Annotated reviews are not available for download in order to protect the identity of reviewers who chose to remain anonymous.

---

## Round 0.3 · accepted · Accept

· Academic Editor

Accept

Reviewers are satisfied with the revisions, and I concur to recommend accepting this manuscript.

Reviewer 2 ·

Basic reporting

no comment

Experimental design

no comment

Validity of the findings

no comment